# Individual Fit Testing of Hearing-Protection Devices Based on Microphones in Real Ears among Workers in Industries with High-Noise-Level Manufacturing

**DOI:** 10.3390/ijerph17093242

**Published:** 2020-05-06

**Authors:** Chien-Chen Chiu, Terng-Jou Wan

**Affiliations:** 1Doctoral Program, Graduate School of Engineering Science and Technology, National Yunlin University of Science and Technology, No. 123 University Road, Section 3, Douliou, Yunlin 64002, Taiwan; 2Department of Safety Health and Environmental Engineering, National Yunlin University of Science and Technology, No. 123 University Road, Section 3, Douliou, Yunlin 64002, Taiwan

**Keywords:** hearing-protection device (HPD), noise exposure, personal attenuation rating (PAR), field microphone in real ear (F-MIRE), pre-workforce education

## Abstract

Hearing-protection devices (HPDs) are particularly important in protecting the hearing of workers. The aim of this study was to prevent hearing damage in workplaces in Taiwan. It was conducted to determine the actual sound attenuation of the personal attenuation rating (PAR) values when wearing HPDs via measurements from field microphones in workers’ real ears (F-MIRE). Across 105 measurement trials for the Classic™ roll-down foam earplug HPDs worn by the workers, there were 23 cases of ineffective protection (including caution and fail); the proportion was 20% (including the first measurement and re-wear of HPDs after education and training). In addition, re-education and training in how to wear the HPDs was provided, improving wearing skills. A total of 29 testees wearing the Classic™ roll-down foam earplug HPDs failed to meet the pass standard for the first PAR test, and 6 of them improved and subsequently passed the PAR test. The improvement rate was 20%. These 23 testees switched to another HPD, namely Kneading-Free Push-Ins™ earplugs. From this group, 16 effective sound attenuation values were obtained, with an improvement rate of 70%. However, seven testees failed to pass the PAR test, and after education, training, and replacement of HPDs with different types, they still could not pass the PAR test. At that time, even if the UltraFit™ pre-molded earplugs were adopted again for wear and replacement, they were still unable to pass the PAR test. This HPD was eventually replaced with the PELTOR X4A Earmuff HPD and then tested again, with these HPDs finally passing the PAR test. In Taiwan, the use of fit testing has been increasing but it is not a common practice, and few studies on hearing-protection fit testing have been conducted in this country. The goal of this study was to gain more insight into the current hearing protection situation, including field attenuation of HPDs obtained by workers, the effects of training on improving the attenuation of HPDs after F-MIRE measurements, and the awareness of hearing health and motivation on the use of HPDs in a high-noise-level environment.

## 1. Introduction

Noise-induced hearing loss (NIHL) is one of the most common health hazards faced by workers in the workplace. According to the U.S. Bureau of Labor Statistics (BLS), from 2004 to 2010, the incidence rate of hearing loss in the manufacturing industry was 12.9%–16.9% per 10,000 workers, which makes it the most common nonfatal occupational injury [1]. In addition, in accordance with the reporting statistics for occupational diseases from the National Diagnosis and Treatment Network for Occupational Diseases and Injuries, the Occupational Safety and Health Administration (OSHA) in Taiwan showed that the number of reported occupational diseases in 2018 (including prevention and treatment centers and network hospitals) was 2158. Among these figures, the highest rate was 813 events of occupational musculoskeletal diseases, accounting for 37.7%, followed by 609 events of occupational hearing loss, accounting for 28.2%; the third highest rate was 329 events of occupational skin disease, accounting for 15.2% [2]. Survey data from the Institute of Labor, Occupational Safety and Health (IOSH) in Taiwan show that the manufacturing industry in Taiwan includes various types of factories, and their noise ranges are recorded as follows: steel (80–110 dBA), textile (80–115 dBA), oil (80–100 dBA), machinery (80–110 dBA), and building (80–115 dBA) [3].

Occupational noise exposure limits are generally based on the assumption of employees working 8 hours a day for 5 days a week. The average sound pressure level for an 8 hour workday, i.e., the daily noise exposure level (or so-called Lex, 8 hours), is measured to determine whether it complies with the law [4]. The basic regulations set out in Taiwanese law are listed in Table 1.

To date, comprehensive studies and reports have assessed hearing protection. Daniell et al. (2002) found that most industries pay little attention to noise control, instead relying on the use of HPDs. Increasing the wearing rate of HPDs depends on the effective HCP [8,9]. Miyauchi et al. (2000) also indicated that up to 29% of workers do not wear HPDs properly [10]. Heyer et al. (2011) showed that the mandatory use of HPDs can help to significantly reduce the hazard of hearing loss caused by noise [11]. Furthermore, there is a low proportion of people wearing HPDs in the industry, or no training provided on how to wear HPDs [12]. Four simplified indices (listed in Table 2) are often used for hearing-protection performance evaluation of HPDs: (1) high, medium, low (HML); (2) single-number rating (SNR); (3) noise-reduction rating (NRR); and (4) octave band (OB) [13]. 

For business entities in Taiwan, most adopt the NRR value of ANSI S12.6 (1984) to assess the hearing-protection performance of HPDs as a basis for selecting HPDs. The NRR, which is a single index of the sound attenuation value, is a tested value measured under pink background noise (total energy is 107.9 dBC) with 100 dBC in each band of energy among the octave noise [14]. There are two international standards for the noise-reduction value of HPDs, SNR and NRR: SNR is the single-value noise-reduction value detected according to the international standard ISO 4969-2; NRR is a single-valued noise-reduction value detected according to the American standard ANSI S3.19-1974. In comparisons of two kinds of HPDs, SNR will be above 3 dB higher than NRR.

NRR considers the individual differences of the wearer. Thus, the calculated protection average is subtracted by two standard deviations to determine the point at which 97% of wearers can wear this HPD and achieve the desired NRR value. Based on the consideration of protecting workers’ hearing from damage due to exposure, institutions should choose HPDs marked with the tested sound-attenuation value according to the conditions of the workplace [15]. Although the NRR aims to achieve a theoretical noise reduction for 98% of workers if they follow the HPD instructions, according to the research, fewer than 5% of workers actually reach the protection level predicted by the NRR [16,17]. At present, there are many systems available for measuring the attenuation performance of HPDs worn by workers in workplaces, including the following: real-ear attenuation at threshold (REAT) (see Method B of ANSI S12.6-1997 and SA/SNZ 1270), which is particularly intended for continuous usage conditions and can provide estimates of field performance [15], field microphone in real ear (F-MIRE), and acoustic text fixture (ATF). In this study, one of these test methods was specifically introduced, the F-MIRE [18] system, which combines a single miniature two-element microphone and relevant professional technologies [19,20,21].

In Taiwan, the labor noise exposure in many industries still exceeds the regulation of 90 dBA for TWA (the time-weighted average sound pressure level during an 8 hour working day with noise exposure). Even if employers provide and require the wearing of HPDs for tasks involving noise, there are still laborers who are classified as “health management level 4” in health management, the cause of which is worthy of in-depth discussion [22,23]. Therefore, this study was conducted through onsite visits in the high-noise manufacturing industry to two steel plants (Factories A and B) and a textile factory (Factory C). The 3M™ EA-RfitTM (3M, St. Paul, MN, USA) HPD adaptability evaluation system is shown in Figure 1. It was adopted to measure the personal attenuation rating (PAR) of laborers while wearing HPDs in these three noisy workplaces. In this study, we also analyzed the spectral characteristics for the high-noise work type and implemented an external sound pressure level test after HPDs were worn. We expected to establish an HPD-fit test method and to determine whether it met the noise exposure standards. We provide an effective evaluation of hearing-protection devices as references.

## 2. Methods

### 2.1. Test Facility and Equipment

The attenuation of HPDs implemented via the F-MIRE test method, which examines the relationship between the insertion loss (IL) and the noise reduction (NR), where TFOE is the transfer function of the open ear, is shown in Figure 2. When the HPD (earplugs or earmuffs) is worn, the exposure volume in the ear is A’, and the noise exposure volume at the position of the external reference point wearing the HPD is B’. The difference between A’ and B’, i.e., B’ – A’, represents the noise reduction (NR). However, regardless of the type of HPD, while subjective sensory testing by the human ear is used, the difference between the sound pressure level without and with the HPD, i.e., A – A’, is called the insertion loss (IL), which is equivalent to the difference in the hearing threshold measured whether the HPD is worn or not, i.e., REAT. Therefore, when the NR and IL values are compared with each other, no correction will result in an error of 5–10 dB in the measured value [22].

The 3M™ EA-RfitTM HPD adaptability evaluation system is composed of software and hardware. Figure 1 provides an image of the microphone and probed earplug tips. The F-MIRE system consists of a sound source that can generate high levels of wideband random noise at the testee’s ear, a dual-element microphone that measures in a quotable location both the sound present at the outside of the earplug and the sound present in the ear canal after having passed through the earplug, a probed earplug to act as a substitute for the actual earplug that subjects will wear, and a robust analysis system installed on a PC laptop that can rapidly take accurate and repeatable measurements in 10 s. The sound levels used, depending upon the level of attenuation provided by the earplug, were up to 90 dBA. The testee’s nose was positioned 40 cm from the front of the loudspeaker. Most 3M™ HPDs can be measured using this evaluation system to obtain their attenuation. The actual measurement only takes approximately 10 s, which can provide the seven standard test frequencies (from 125 Hz to 8 KHz) and the individual binaural PAR values.

In this study, 50 male workers from noisy work areas of steel mills (A, B) and textile factories (C) served as the testees for testing. We fully explained the research to all testees, and the research was approved by the occupational safety and health management entities of the three participating factories and was conducted with the consent of all testees.

### 2.2. Types of HPDs

Various types of HPDs were prepared in the factories participating in this research so that the workers could select the appropriate type. A total of four different types of HPD were adopted in this study (as listed in Table 3): 3M™ Classic™ (3M, St. Paul, MN, USA) roll-down foam earplugs, 3M™ Push-Ins™ (3M, St. Paul, MN, USA) stemmed-style pod plugs, 3M™ UltraFit™ (3M, St. Paul, MN, USA) pre-molded earplugs, and 3M™ PELTOR X4A Earmuff (3M, St. Paul, MN, USA). In addition, the number of experimental trials using these four types of HPD was 111, 23, 7, and 7, respectively.

### 2.3. Experimental Procedure

The research trials were conducted in the office areas of each participating factory in this study. The background noise in the office was less than 80 dBA, complying with the regulations of the 3M™ EA-RfitTM HPD adaptability evaluation system. The evaluation process is shown in Figure 3; at the beginning, during the untaught condition, researchers observed the testees’ HPD-wearing skills from the side and used these results as the benchmark for the wearability of the HPD. If the PAR value did not meet the pass standard, i.e., did NOT meet the requirement (Figure 4), or if an inappropriate manner of wear was observed, even if the testee’s PAR value met the passing standard, the researchers taught the testee how to properly wear the HPDs and then repeated the adaptability test. After the teaching component, if the PAR value still indicated a fail or showed no signs of improvement, a different type of HPD and additional instructions were provided, and the adaptability was tested again with the new HPD until the PAR value met the “pass” standard (Figure 5). The evaluation steps were as follows:After each testee entered the test site, the engineers first explained the PAR test procedure;The testee wore the earplugs with probes (Classic™ roll-down foam earplugs);The researchers connected the two-element miniature microphone to the left and right probes in the testee’s HPD;The testee faced the loudspeaker at the designated position, keeping the ear and loudspeaker at the same height, and the distance from the nose to the clip on the loudspeaker was 40 cm (Figure 6);The testee’s PAR value was obtained using the F-MIRE test method;The binaural PAR dB was the difference between the binaural PAR dB and the binaural variability; andIf the sum of the binaural PAR dB and the statutory noise value in the workplace exceeded the setting standard for participating research factories, the result was a pass; otherwise, the result was a fail.

## 3. Results

In this study, a total of 148 PAR tests were conducted for Factories A, B, and C. Of these, 52 were held in Factory A, where the testees wore the HPDs sequentially as follows: the Classic ™ roll-down foam earplugs 36 times, accounting for 69%; the Push-Ins™ stemmed-style pod plugs 12 times, accounting for 23%; the UltraFit™ pre-molded earplugs 2 times, accounting for 4%; and the PELTOR X4A Earmuff 2 times, accounting for 4%. Factory B followed with 23 testees, accounting for 16%. The testees wore the HPDs sequentially as follows: the Classic ™ roll-down foam earplugs 18 times, accounting for 78%; the Push-Ins™ stemmed-style pod plugs 4 times, accounting for 17%; and the UltraFit™ pre-molded earplugs 1 time, accounting for 5%. In addition, Factory C was the site of 73 tests, accounting for 49%. The testees wore the HPDs sequentially as follows: the Classic ™ roll-down foam earplugs 57 times, accounting for 78%; the Push-Ins™ stemmed-style pod plugs 7 times, accounting for 10%; the UltraFit™ pre-molded earplugs 4 times, accounting for 5%; and the PELTOR X4A Earmuff 5 times, accounting for 7%. Statistical data for the frequencies and percentages of the abovementioned four various types of HPD for all 148 PAR tests across the three different factories are shown in Table 4.

In addition, of the PAR tests for the four HPDs used in this study, the largest number was conducted for the Classic ™ roll-down foam earplugs (up to 105 times, accounting for 71%). The Push-Ins™ stemmed-style pod plugs followed with 23 times, accounting for 16%. The UltraFit™ pre-molded earplugs were tested 7 times, accounting for 5%, and the PELTOR X4A Earmuff was tested 7 times, accounting for 5% (Table 4).

The different environments of Factories A, B, and C may have affected the PAR values, as summarized in Table 5, which lists the relevant statistical analysis and test data for the PAR values from the three participating factories in this study. In Factory A, the mean (M) value of the left PAR dB was 25.5, and the standard deviation (SD) value was 8.3. The M value of the right PAR dB was 25.9 (SD = 6.9). For the binaural PAR dB, the M value was 22.7 (SD = 7.3). In Factory B, the M value of the left PAR dB was 24.7 (SD = 8.8); the M value of the right PAR dB was 22.2 (SD = 10.6); and the M value of the binaural PAR dB was 19.6 (SD = 9.9). In Factory C, the M value for the left PAR dB was 27.4 (SD = 8.4); the M value of the right PAR dB was 25.2 (SD = 9.3); and the M value of the binaural PAR dB was 22.7 (SD = 9.3). Furthermore, one-way ANOVA was used in the statistical test approach for the three participating factories in this study, yielding the following results for the left PAR dB: F = 4.192, *p* = 0.018 < 0.05. These findings indicated that the F-value for left PAR dB was 4.192, and the *p*-value was 0.018, which was less than 0.05 (the level of significance). Similarly, the other findings were as follows: right PAR dB: F = 4.120, *p* = 0.019 < 0.05; binaural PAR dB: F = 3.625, *p* = 0.030 < 0.05. Thus, all three variables showed significant differences. Another post hoc test via Scheffe’s method showed that for the left PAR dB, C > A, indicating that Factory C had a higher value than Factory A. Similarly, for the right PAR dB, C > B (Factory C was higher than Factory B), and for the binaural PAR dB, C > B (Factory C was higher than Factory B).

Additionally, across the 111 measurement trials for the Classic™ roll-down foam earplugs worn by the workers (Table 6), there were 29 cases of ineffective protection (including caution and fail), a proportion of 22% (including the first measurement and re-wear of HPD after education and training). In addition, via the re-education and training of the wearing method to improve wearing skills, 23 testees wearing the Classic™ roll-down foam earplug HPDs failed to meet the pass standard for the first PAR test, and 6 of them improved and subsequently passed the PAR test. The improvement rate was 20%. These 23 testees then switched to another HPD, namely Kneading-Free Push-Ins™ earplugs; at this time, 16 effective sound attenuation values were obtained, and the improvement rate was 70%. However, seven testees failed to pass the PAR test after education training and the replacement of different types of HPD and could not pass the PAR test. In these cases, even if the UltraFit™ pre-molded earplugs were worn and replaced again, they were still unable to pass the PAR test. They were eventually replaced with the PELTOR X4A Earmuff HPDand then tested again; this HPD finally passed the PAR test. The analysis results revealed that wearing Classic™ roll-down foam earplugs was the most convenient, lowest cost, and protective choice for 78% of the testees. Among these, four testees did not achieve hearing protection because of ear canal structural problems, and could not be protected until their HPDs were replaced with earmuffs.

## 4. Discussion

In this study, three high-noise job site visits were completed, and PAR measurement evaluations were conducted 140 person-times. During the process of the onsite visits, it was also found that after factors such as economy, convenience, and personnel acceptance were considered, most of the business entities still used mainly earplug-type HPDs. Only a few workers in specific working environments used earmuff-type HPDs. The general conclusions are summarized as follows:The highest usage rate of the earplug-type HPDs was the Classic ™ roll-down foam earplugs. For employers, this was because of their relatively low price and convenience for workers to wear.The F-MIRE test method could separately measure the sound pressure inside and outside the ear after HPDs were worn, which is intuitive and convenient. However, the test result was easily affected by incorrect posture and contact with the inner wall of the ear canal during wear. For the same type of HPD at different work sites, the measured average PAR value would also be different (Table 4).The test result under professional instructions and re-wear of HPD for the testees could indeed show improved protective efficiency of HPD for the wearers (excluding those with different ear canals, which could increase by ca. 35%). This confirmed the effectiveness of education and training.At present, the commercially available HPD adaptability test and evaluation system is mainly based on the individual test applications of earplugs and earmuffs. Herein, the F-MIRE test method, among the systems adopted in this study, was convenient to implement at noisy work sites, and the evaluation results could be quickly obtained.This study was limited by the fact that it could only be performed during the production period of each plant, and the experimental sampling needed to be completed within a limited time. This study is the first to use the commercially available hearing protectors with the highest usage rate in the current industry. During this period, 105 samples were taken.

## 5. Conclusions

Based on the results and conclusions of related literature and field analysis, the following suggestions are proposed for future research:The F-MIRE test method could be utilized as an educational training tool to help workers to wear and select HPDs, as it can directly evaluate the protective efficiency of HPDs for wearers. However, attention should be paid to the ambient background noise when the test is conducted.The current F-MIRE test method uses a miniature microphone and its extended ear canal probe to measure the sound pressure inside the ear in the external ear canal after earmuffs are worn. This method can directly evaluate the protective efficiency as individuals wear the HPD, but its ear canal probe tube easily touches the inner wall of the ear canal which affects the measurement results. Relevant improvements in measurement methods could be further explored in the future.The sound attenuation value, or NRR value, which is marked on the HPD, did not truly reflect the effects of workers wearing HPDs at noisy work sites, and its value was not higher. The best HPD was suitable for the wearers; they could communicate without any obstacles when worn and it could be used effectively for a long time, i.e., it could effectively perform its function and be used continuously.This study found that some test subjects had smaller ear canals and the polymeric partially compressible foam earplugs were not easy to insert, making the workers’ earplugs likely to fall off when worn for a long time. Thus, these workers should consider replacing their earbuds with different styles.In future work, we will assign different forms of hearing protection to the same tester to verify the best type and performance of personal hearing protection.

## Figures and Tables

**Figure 1 ijerph-17-03242-f001:**
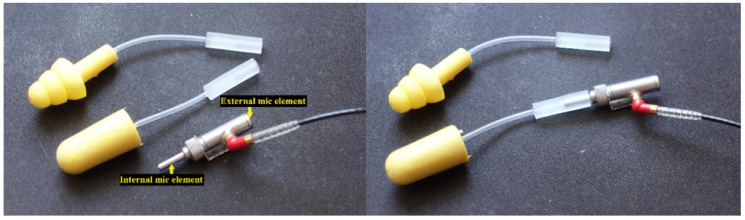
The equipment used in the F-MIRE test method in this study.

**Figure 2 ijerph-17-03242-f002:**
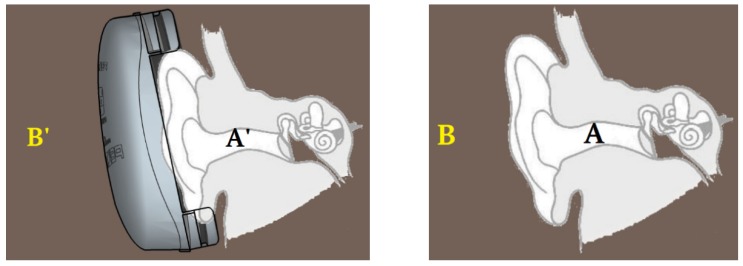
Schematic diagram for the measurement location in the evaluation of the attenuation of HPDs via the F-MIRE test method. ✽ Remarks: insertion loss (IL) = A − A′ ≌ REAT; noise reduction (NR) = B′ − A′; TFOE is measured with respect to Point B, which is generally the head-center location with the head absent. NR = IL − TFOE.

**Figure 3 ijerph-17-03242-f003:**
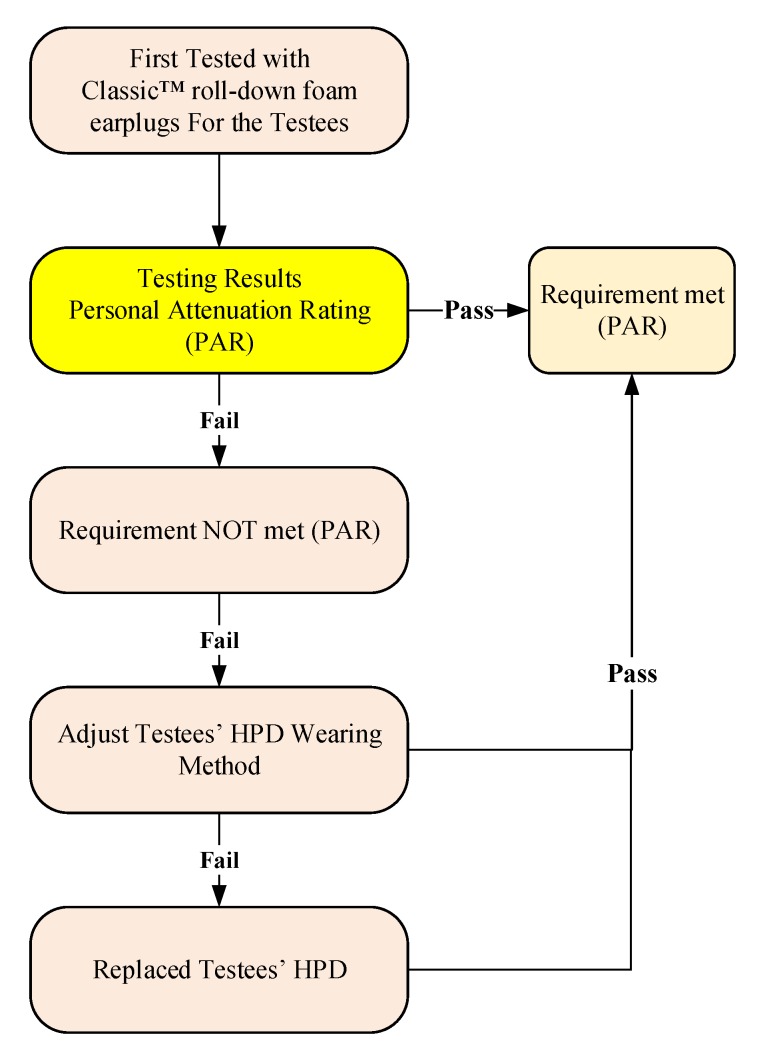
The flowchart of the evaluation process for the 3M™ EA-RfitTM HPD adaptability evaluation system applied in this study.

**Figure 4 ijerph-17-03242-f004:**
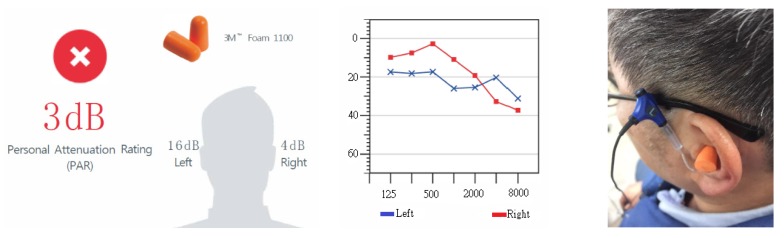
An example where the test result of the PAR value was “fail” in the first test in the untaught situation of the testee.

**Figure 5 ijerph-17-03242-f005:**
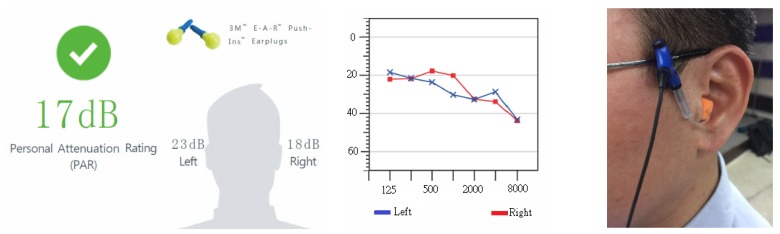
An example where the test result of the PAR value was “pass” after additional instructions and a different type of HPD were provided, followed by a re-test of the adaptability with the new HPD.

**Figure 6 ijerph-17-03242-f006:**
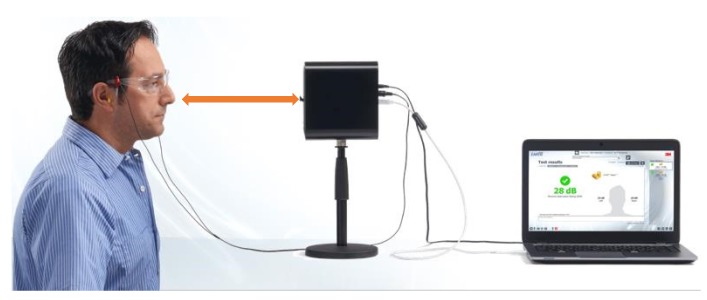
The distance from the testee’s nose to the clip on the loudspeaker was 40 cm.

**Table 1 ijerph-17-03242-t001:** The Taiwanese regulations on hearing protection.

Regulations	Regulatory Requirements
Occupational Safety and Health Act	Employers shall have the necessary safety and health equipment and measures that comply with regulations to “prevent the risks of injuries posed by radiation, high temperature, low temperature, ultrasonic waves, noise, vibration, and abnormal atmospheric pressure” [5].
Enforcement Rules of the Occupational Safety and Health Act	States expressly that job site monitoring plans shall be formulated with the implementation of monitoring whether the job sites are emitting extreme noise [6].
Occupational Safety and Health Facilities Regulation	Claims that 85 dBA is the action level, and the hearing conservation program (HCP) must be implemented as required. In workplaces where the time-weighted average sound pressure level for an 8 hour workday (8 hour TWA) exceeds 85 dBA or the exposure dose exceeds 50%, the employers shall have the following hearing protection measures taken, and an execution record shall be made and retained for three years: Noise monitoring and exposure assessment; Noise hazard control; Selection and wearing of hearing-protection devices (HPDs); Hearing-protection education and training; Management and examination of health; and Effectiveness evaluation and improvement [7].

**Table 2 ijerph-17-03242-t002:** Hearing-protection performance evaluation of hearing-protection devices (HPDs).

Method	Illustration/Formula
high, medium, low (HML)	Referring to ISO 4869-2 (1992), three values are provided to calculate the sound attenuation values of high frequency (H), intermediate frequency (M), and low frequency (L).
	(1) ${\;H}_{x}=0.25\overset{4}{\underset{i=1}{\sum }}{\mathrm{PNR}}_{\mathrm{xi}}-0.48\overset{4}{\underset{i=1}{\sum }}(\mathrm{di}.{\mathrm{PNR}}_{\mathrm{xi}})$ (2) ${\;M}_{x}=0.25\overset{8}{\underset{i=5}{\sum }}{\mathrm{PNR}}_{\mathrm{xi}}-0.16\overset{8}{\underset{i=5}{\sum }}(\mathrm{di}.{\mathrm{PNR}}_{\mathrm{xi}})$ (3) ${\;L}_{x}=0.25\overset{8}{\underset{i=5}{\sum }}{\mathrm{PNR}}_{\mathrm{xi}}-0.23\overset{8}{\underset{i=51}{\sum }}(\mathrm{di}.{\mathrm{PNR}}_{\mathrm{xi}})$
single-number rating (SNR)	Referring to ISO 4869-2 (1992) for a single index of sound attenuation value. It is the value measured under the pink background noise of 100 dB (C).
	${\mathrm{SNR}}_{xi}=100\mathrm{dBC}-101\log\overset{8}{\underset{K=1}{\sum }}10^{0.1\;\left(\;\mathrm{LAf}\left(K\right)-\mathrm{APVf}\left(k\right)\;\right)\;},\mathrm{dB}\left(A\right)$
noise-reduction rating (NRR)	Referring to ANSI S12.6 (1984) for a single index of sound attenuation value. It is the value obtained by testing under the pink background noise (total energy 107.9 dBC), in which the energy of each frequency band in the eight band noise is 100 dB (C).
	${\mathrm{NRR}}_{}=107.9\mathrm{dBC}-101\;\log\overset{8}{\underset{K=1}{\sum }}10^{0.1\;\left({\;\mathrm{LA}}_{f\left(k\right)}-{\mathrm{APV}}_{f\left(k\right)}\;\right)\;}-3\mathrm{dB}\left(A\right)$
octave band (OB)	Referring to ISO 4869-2 (1992), after calculating the sound attenuation value of the eight note frequency band separately, the sound attenuation values of the HPDs can be obtained.
	${L^{\prime }A}_{}=101\;\log\overset{8}{\underset{K=1}{\sum }}10^{0.1\;\left({\;L}_{f\left(k\right)}+A_{f\left(k\right)}-{\mathrm{APV}}_{f\left(k\right)}\;\right)\;}\mathrm{dB}\left(A\right)$

**Table 3 ijerph-17-03242-t003:** Various types and appearances of the HPD samples tested in this study.

Type.	Classic™ Roll-Down Foam Earplugs	Push-Ins™ Stemmed-Style Pod Plugs	UltraFit™ Pre-Molded Earplugs	PELTOR X4A Earmuff
**Appearance**	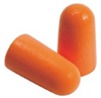	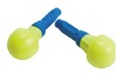	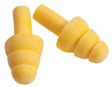	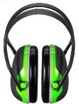

**Table 4 ijerph-17-03242-t004:** Distributions of the 148 PAR tests and percentages of the four types of HPDs at each factory.

		Classic™ Roll-Down Foam Earplugs	Push-Ins™ Stemmed-Style Pod Plugs	UltraFit™ Pre-Molded Earplugs	PELTOR X4A Earmuff	Total
**Factory**	**A**	36 (69%)	12 (23%)	2 (4%)	2 (4%)	52
	**B**	18 (78%)	4 (17%)	1 (5%)	0 (0%)	23
	**C**	57 (78%)	7 (10%)	4 (5%)	5 (7%)	73
**Total**		**1** **11**	**23**	**7**	**7**	**14** **8**

**Table 5 ijerph-17-03242-t005:** Relevant results of statistical analysis and tests of the PAR values from the three participating factories in this study.

Factory		A	B	C	*F* value	Significance (*p* value)	Scheffe’s Method
**Left PAR dB**	Mean (*M*)	25.5	24.7	27.4	4.192	0.018 *	C > A
	Standard deviation (*SD*)	8.3	8.8	8.4			
**Right PAR dB**	Mean (*M*)	25.9	22.2	25.2	4.120	0.019 *	C > B
	Standard deviation (*SD*)	6.9	10.6	9.3			
**Binaural PAR dB**	Mean (*M*)	22.7	19.6	22.7	3.625	0.030 *	C > B
	Standard deviation (*SD*)	7.3	9.9	9.3			

* *p* < 0.05.

**Table 6 ijerph-17-03242-t006:** Distributions of the PAR test results and percentages of the four types of HPD.

Classic™ Roll-Down Foam Earplugs			Push-Ins™ Stemmed-Style Pod Plugs	UltraFit™ Pre-Molded Earplugs	PELTOR X4A Earmuff
1st test		2nd test			
**Caution**	10 (4%)	4 (14%)	-	4 (57%)	-
**Fail**	19 (18%)	19 (66%)	7 (30%)	3 (43%)	-
**Pass**	82 (78%)	6 (20%)	16 (70%)	-	7 (100%)
**Total**	**111 (100%)**	**29 (100%)**	**23 (100%)**	**7 (100%)**	**7 (100%)**

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
