# Peer review of "Individual Fit Testing of Hearing-Protection Devices Based on Microphones in Real Ears among Workers in Industries with High-Noise-Level Manufacturing"

_ijerph, 2020, doi:10.3390/ijerph17093242_

Round 1
Reviewer 1 Report
Well-conducted and well-described study.
it may be worth adding a few words to the abstract or early in the introduction about the fact that the study was conducted in Taiwan, which explains the focus on Taiwan in the introduction. Otherwise the first mention is in the last paragraph of the introduction.
The last sentence of the introduction is not clear.
Minor improvements in English language expression required.
Line 150 testee's
Figure 2 - the text inside the left-image is not readable in my copy.
A statistical analysis compares the factories but there is no insight into the discussion as to why this analysis is useful or what underlies the difference.
Author Response
Dear Prof.
Peace be with you and your family. Attached please find a well-prepared remanuscript of our research associated with the field of International Journal of Environmental Research and Public Health, entitled “Assessment of the effect of training along with fit testing on" Individual fit testing of hearing protection devices based on microphone in real ear among workers in industries with high noise level manufacturing”.
Actually, the authors, we have done a great effort to enrich the completeness and applicability of this paper, and continuously to improve the quality and quantity of our research for meeting the needs of “International Journal of Environmental Research and Public Health” Journal as possible as we can. The full-paper has been composed appropriately, and has been self-reviewed
constantly and repeatedly in accordance with the Aims and Scope and with correct format.
In addition, we have done our best for the submission of this research all thoroughly done, and sincerely wish to have the opportunity to publish for the Journal Article at “International Journal of Environmental Research and Public Health”, and that you consider the contribution favorably.
If something needs to be done further, please feel free to contact me. We look forward to hearing from you. Thank you very much.
Thanks for your valuable and great help!
Sincerely yours,
Corresponding Author
Chien-Chen Chiu

Reviewer 2 Report
Thank you for submitting this manuscript. My comments are mostly stylistic rather than content per se. You may want change your title given the stated aim mentioned as early as in the abstract. If the aim is really to prevent damage from noise exposure in the work place. Fit testing is important, but it is only one of several elements that contribute to hearing loss prevention.
Here are my line by line comments:
line 30: change word "of" to "with"
line 56: delete "so-called LEX" and instead use "Lex"
line 54-76: This section on Taiwan law is complicated and may not be of interest to the readers. You may want to put this section either in an appendix or put it in a chart.
line 78: In references and this comment is for the entire manuscript.... Add date for Daniell et al. (ADD YEAR). In this case it is 2002, but the format should always have the date.
line 79: Change to proper citation eg. Miyauchi et al. ( ) rather than just jumping in to say "Miyauchi's research...."
line 80: Heyer et al. ( ). etc.....
line 83: start a new paragraph at "Four simplified..."
lines 83-85: expand the description further on each of the methods, or at least show them in chart form.
line91: explain why sound attenuation value should be reduced by 3 dB.
line 94: Actually the NRR contains standard deviation measures that are subtracted from actual laboratory attenuations in order to simulate real life usage but the way you wrote it, you implied that the NRR would still be higher- are the two standard deviation subtractions not sufficient, and if not, you should discuss this.
line 114: Expand on the description of the 3M EA-RfitTM system (at least say "see figure 2").
line 208: change the text of figure 2 from "ca" to "circa" or delete it.
line 209: I found the results section to be very confusing. Perhaps put the results in a summary chart form. You seem to just repeat in the text what was in the chart and this was redundant- a summary chart would be better.
line 250: Your data in table 4 was really only borderline significant (between 0.01 and 0.05) but with the post hoc tests, the power would be reduced, so the data were really right on the line of statistical significance.
line 296: Suggest deleting the sentence "If any hearing protection standards are not met, other types...." should be left out. Its not really a standard that you want to see being met; its the frequency by frequency attenuation that should be below a critical level such as 85 dBA.
Finally consider using the proper phrase throughout of "polymeric partially compressible foam earplug" rather than "rubbing type".
Author Response

(The authors gave the same response as above.)
